# High BMI-attributed diabetic kidney disease in China versus globally, 1990–2021, with projections to 2045: Divergent trends and an accelerating burden

Luyao Jiang[1], Jinkang Jia[2], Tianxi Chen[3]*, Linlin Xu[3], Miaojun Shi[3]

**1** Department of Rehabilitation Medicine, The First People's Hospital of Yongkang, Jinhua, Zhejiang, China, **2** Department of Nephrology, The Second Affiliated Hospital of Zhejiang University, Hangzhou, Zhejiang, China, **3** Department of Nephrology, The First People's Hospital of Yongkang, Jinhua, Zhejiang, China

* ykrmctx@163.com

## Abstract

### Objective

To analyze the accelerating burden and unique trends of diabetic kidney disease (DKD) attributable to high body mass index (BMI) in China from 1990 to 2021, contextualized against global patterns.

### Methods

Relevant data were extracted from the Global Burden of Disease 2021 study. Trends in disease burden were analyzed using a joinpoint regression model. The disease burdens during the period of 2022–2045 were forecasted using the NORDPRED model.

### Results

From 1990 to 2021, the absolute burden of high BMI-attributable DKD increased dramatically in China, with the number of deaths surging by 355%. While global trends also rose, China exhibited unique characteristics: a disproportionately high burden was observed in men aged 85 and above, a pattern not seen globally. Spatiotemporal analysis revealed China progressed to a higher global burden stratum over the three decades. Projections indicate the number of deaths and DALYs in China will continue to rise steeply through 2045.

### Conclusion

The burden of DKD attributable to high BMI has increased at an alarming rate in China, with distinct demographic vulnerabilities. These findings underscore the

**Data availability statement:** The data used in this study can be accessed at the following website: https://vizhub.healthdata.org/gbd-results/.

**Funding:** This work was supported by the Science and Technology Project of Yongkang City (Grant No. 202212) to TXC and the Science and Technology Project of JinHua City (Grant No. 2025-4-292) to LLX. The funders had no role in study design, data collection and analysis, decision to publish, or preparation of the manuscript.

**Competing interests:** The authors have declared that no competing interests exist.

urgent need for targeted national public health interventions to mitigate this formidable challenge.

## 1. Introduction

With the global obesity epidemic intensifying, the highly populous nation of China is facing an acute crisis. In 2021, over 402 million adults aged 25 years and above in China were classified as overweight or obese, a figure ranking first globally [1,2]. The current international standard for assessing obesity is based on the body mass index (BMI). The World Health Organization classifies individuals with a BMI in the range of 25–30 kg/m2 as overweight and those with a BMI exceeding 30 kg/m2 as obese [1]. Notably, in addition to being a pivotal indicator of obesity, high BMI is a critical risk factor in the development and progression of diabetic kidney disease [3]. DKD is a clinically defined chronic complication of diabetes, characterized by persistent albuminuria (typically measured as a urinary albumin-to-creatinine ratio > 30 mg/g) and/or a progressive decline in kidney function (manifested as a sustained estimated glomerular filtration rate (eGFR) below 60 ml/min/1.73 m2). These clinical signs typically emerge after a period of diabetes duration, and its diagnosis is made clinically without the need for a kidney biopsy, unless other causes of kidney disease are suspected [4]. (DKD)—the primary microvascular complication of diabetes,DKD has emerged as a major threat to the health and life expectancy of the global and Chinese populations [5,6]. To achieve precise disease prevention and control, researchers must thoroughly investigate the changing patterns and predictive trends of DKD burden attributable to high BMI from 1990 to 2021. However, existing studies predominantly focus on single geographic regions or isolated evaluation metrics, lacking a systematic comparative analysis of long-term trends in DKD burden globally and within China.

This research delves into multidimensional data from 1990 to 2021, analyzing the trends in DKD burden attributable to high BMI. This study comprehensively assesses DKD burden across different regions, populations, and socioeconomic strata, both globally and in China. Furthermore, it predicts the DKD burden attributable to high BMI over the period of 2022–2045. This effort is intended to assist governmental bodies and research institutions in formulating refined public health interventions.

## 2. Materials and methods

### 2.1. Data sources

The data for this study were sourced from the publicly available data from the Global Burden of Disease study 2021 (GBD 2021), which provides disease burden data for 204 countries and regions worldwide. GBD 2021 employed a comparative risk assessment framework, quantifying the impact of specific risk factors on population health by comparing their actual exposure levels to a counterfactual scenario defined by the theoretical minimum risk exposure level, while maintaining constant exposure levels of other independent risk factors [7]. GBD 2021

specified the theoretical minimum risk exposure level for high BMI to be 20–25 kg/m2 [7]. On the basis of the literature, this study defines high BMI as BMI > 25 kg/m2 [8]. The study extracted data on deaths and disability-adjusted life years (DALYs) associated with DKD attributable to high BMI in both global and Chinese populations from 1990 to 2021. Age-standardized mortality rate (ASMR) and age-standardized disability rate (ASDR) were calculated using the World Standard Population age structure for trend analysis, and a joinpoint regression model was employed to compute the average annual percentage change (AAPC). This study did not involve any ethical issues because it did not collect any personal information and solely conducted secondary analysis on publicly available data from GBD 2021.

According to the guidelines of the International Classification of Diseases, Tenth Revision, chronic kidney disease attributed to type 1 diabetes in the GBD 2021 data encompasses diseases coded as E10.2–E10.29, while chronic kidney disease attributed to type 2 diabetes encompasses diseases coded as E11.2–E11.29 [9].

### 2.2. Research methods

The evaluation metrics used in this study were number of deaths, mortality rates, DALYs, DALY rates, sociodemographic index (SDI), and ASMR and ASDR along with their spatial and population distributions. DALY, a pivotal indicator of comprehensive disease burden, thoroughly reflects the impact of a disease on population health, with higher values indicating heavier disease burdens [10]. Through standardization, ASMR and ASDR eliminate disparities among different regions and populations, facilitating comparative analysis and trend evaluation [11]. Jointpoint regression analysis was employed to calculate the annual percentage change (APC) and AAPC in ASMR and ASDR. APC is a measure of the annual variation in disease burden, indicating the proportional change in burden relative to the previous year [12]. AAPC is a measure of the average annual growth or decline in the burden of a specific disease or health issue over a defined period [13]. The GBD collaborative group categorizes SDI into five zones: low (SDI < 0.466), low-middle (SDI = 0.466–0.619), middle (SDI = 0.620–0.712), high-middle (SDI = 0.713–0.810), and high (SDI > 0.810). SDI values ranging from 0 to 1 represent the theoretical spectrum from the lowest to the highest developmental levels, with higher SDI values corresponding to affluent nations with more advanced healthcare systems [14].

To assess disease burden, GBD 2021 employed the advanced Bayesian regression tool DisMod-MR2.1 for analysis, modeling, and estimation. To ensure the validity of cross-temporal comparisons of disease burden, the study utilized the World Standard Population age structure for age standardization [15].

### 2.3. Statistics

This study employed the "dplyr" package in R for cleaning the data downloaded from the datasets, the "easyGBDR" package in R for model implementation, and the "ggplot2" package in R for visualization.

## 3. Results

### 3.1. Rapid rise in high BMI attributed DKD burden in China and globally

From 1990 to 2021, the number of deaths associated with DKD attributable to high BMI increased from 40,479–173,263 in the world and from 8,208–37,355 in China (Fig 1A). The ASMR escalated from 1.16 to 2.08 per 100,000 in the world and from 1.26 to 1.94 per 100,000 in China (Fig 1B). The number of DALYs surged from 1,197,972–4,323,077 person-years in the world and from 245,323–941,988 person-years in China (Fig 1C). The ASDR increased from 30.82 to 50.14 per 100,000 in the world and from 30.29 to 45.03 per 100,000 in China (Fig 1D). Compared with 1990, the number of deaths, ASMR, DALYs, and ASDR in 2021 had increased by 328%, 79.31%, 261%, and 62.69%, respectively, in the world and by 355%, 53.97%, 284%, and 48.66%, respectively, in China.

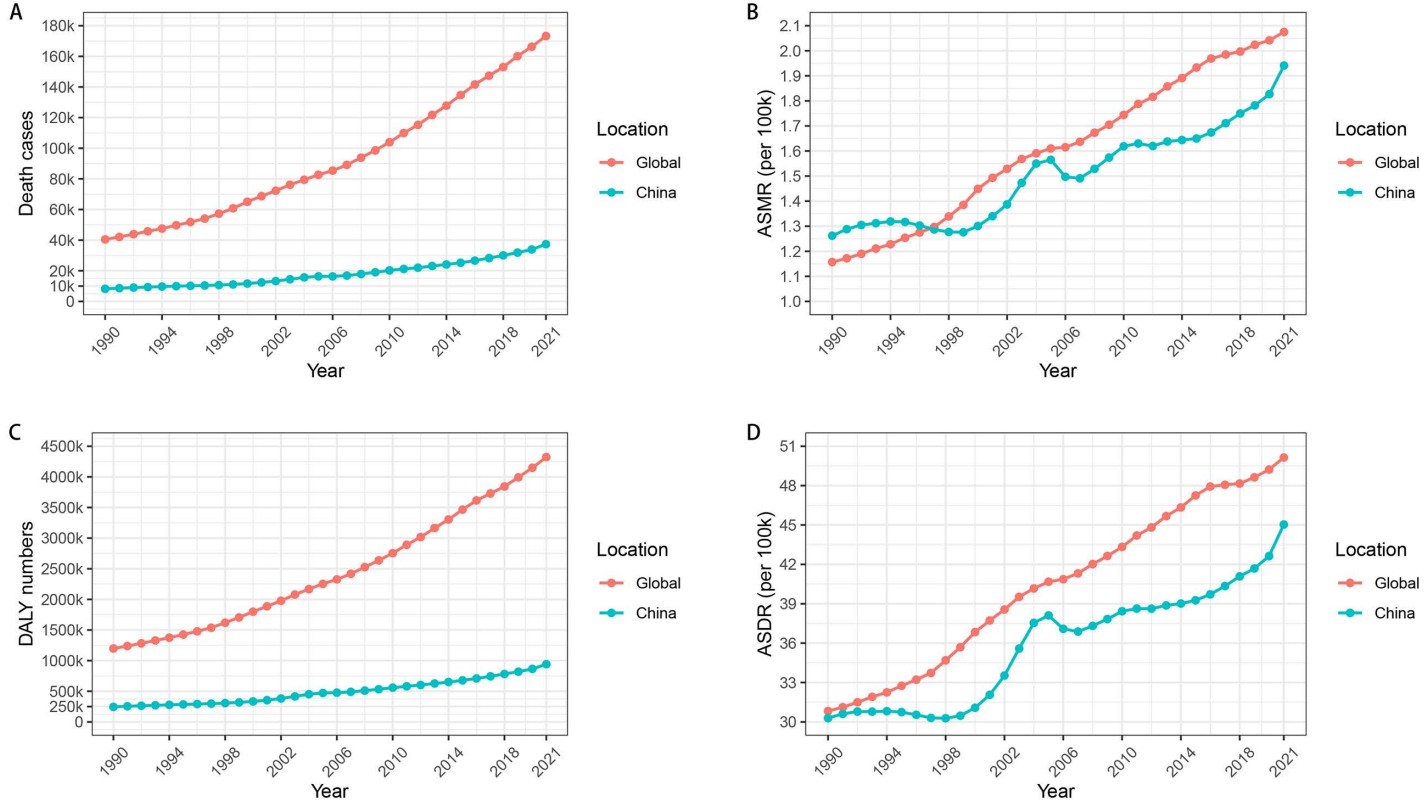

**Fig 1. Global and Chinese burden of DKD attributable to high BMI from 1990 to 2021. (A)** Number of deaths. **(B)** ASMR. **(C)** Number of DALYs. **(D)** ASDR. DKD, diabetic kidney disease; BMI, body mass index; ASMR, age-standardized mortality rate; DALYs, disability-adjusted life years; ASDR, age-standardized disability rate.

### 3.2. Higher proportion of elderly men in the high BMI attributed DKD burden

The analysis of global mortality data revealed that virtually no deaths occurred because of DKD attributable to high BMI among individuals under 25 years of age. The mortality rate began to increase with age among individuals aged over 50 years, and it surged rapidly among those aged 85 years and above. Similar trends in mortality rate with respect to age were observed in China (Fig 2A). Regarding the global disability burden, individuals under the age of 25 years exhibited no DALY loss. The DALY rate gradually increased with age among individuals aged over 50 years and sharply increased among those aged 85 years and above. Similar trends in DALY rates with respect to age were observed in China (Fig 2B). Regarding gender differences, the trends in global mortality and DALY rates resembled an inverted pyramid, with a relatively balanced gender distribution. In China, however, the mortality and DALY rates among men aged 85 years and above were significantly higher than those of their female counterparts. This male-dominated disease burden characteristic was particularly pronounced in the age group of 90–94 years (Fig 2A and 2B).

### 3.3. Unique APC characteristics of ASDR and ASMR

From 1990 to 2021, both the ASMR and ASDR of DKD attributable to high BMI exhibited a significant upward trend globally, with AAPCs of 1.90% and 1.58%, respectively. The ASMR increased especially steeply during the periods of 1997–2000 (APC = 3.79%) and 2000–2003 (APC = 2.84%, Fig 3A). The ASDR displayed a more pronounced upward trajectory from 1996 to 2003 (APC = 2.64%, Fig 3B). From 1990 to 2021, the ASMR and ASDR of DKD attributable to high BMI also

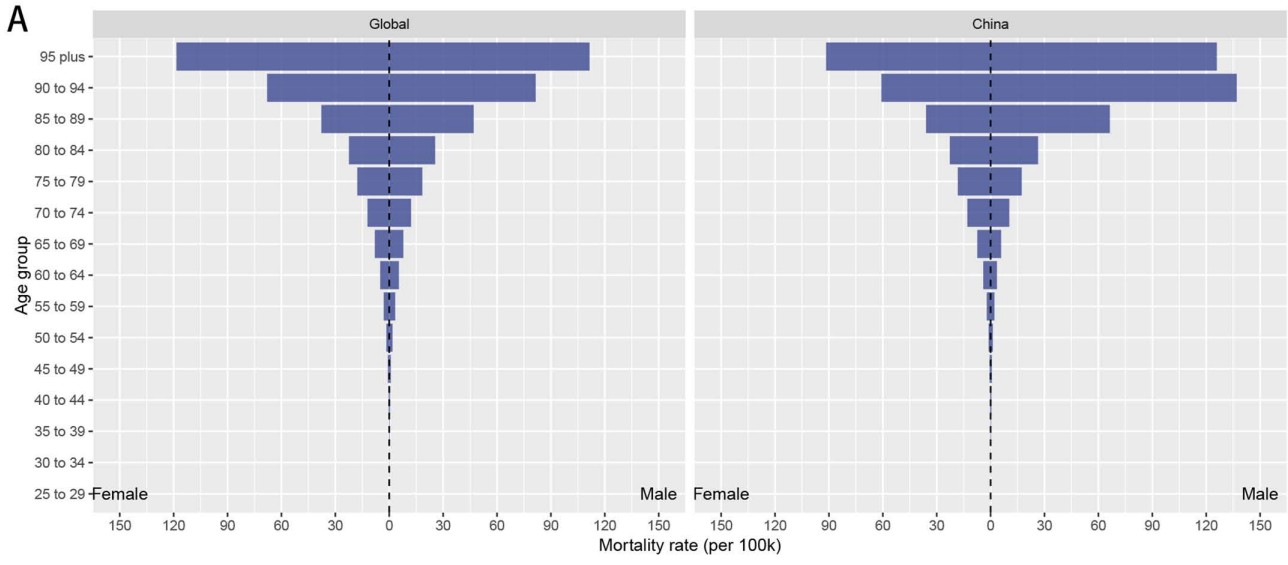

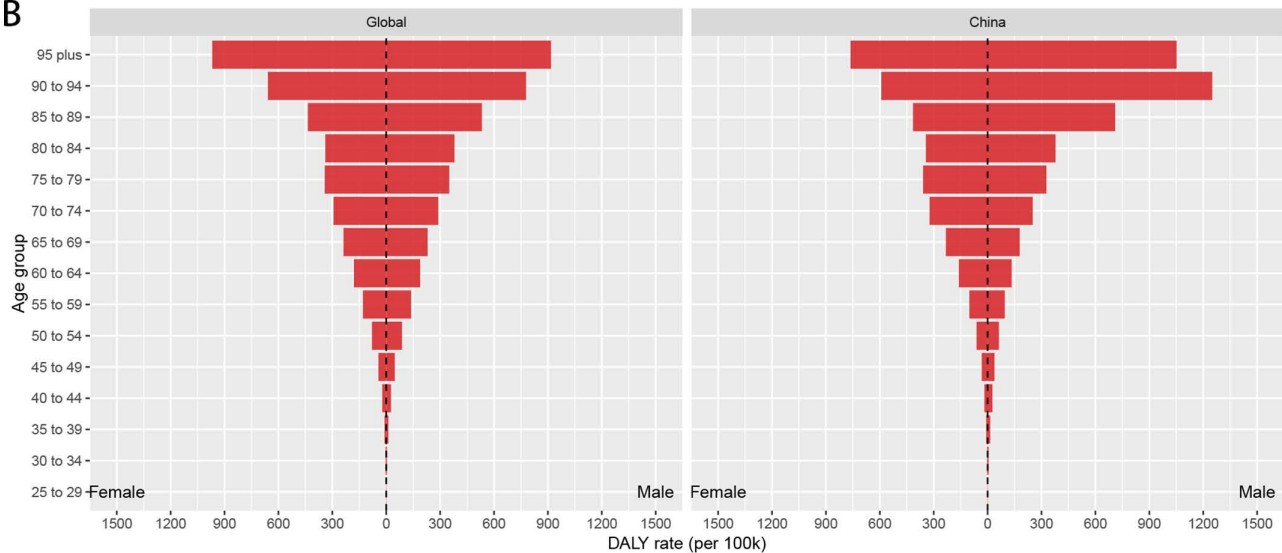

**Fig 2. Global and Chinese age- and sex-specific burden of DKD attributable to high BMI. (A)** Trends in mortality rates across age groups. **(B)** Trends in DALY rates across age groups.

displayed an overall upward trend in China, with AAPCs of 1.25% and 1.22%, respectively. The ASMR grew robustly during the periods of 2000–2004 (APC = 4.68%) and 2016–2021 (APC = 2.81%, Fig 3C). Similarly, the ASDR increased steeply during the periods of 2000–2004 (APC = 5.33%) and 2018–2021 (APC = 3.23%, Fig 3D). Notably, the rising trajectories of the ASMR and ASDR curves were highly congruent both globally and within China.

### 3.4. Spatial distribution patterns of ASDR and ASMR in China and globally

In 1990, the maximum ASMR of DKD attributable to high BMI across the globe was below 11.45 per 100,000 individuals. The Caribbean and Central America and the Persian Gulf region exhibited the highest rates (1.89 to <11.45 per 100,000

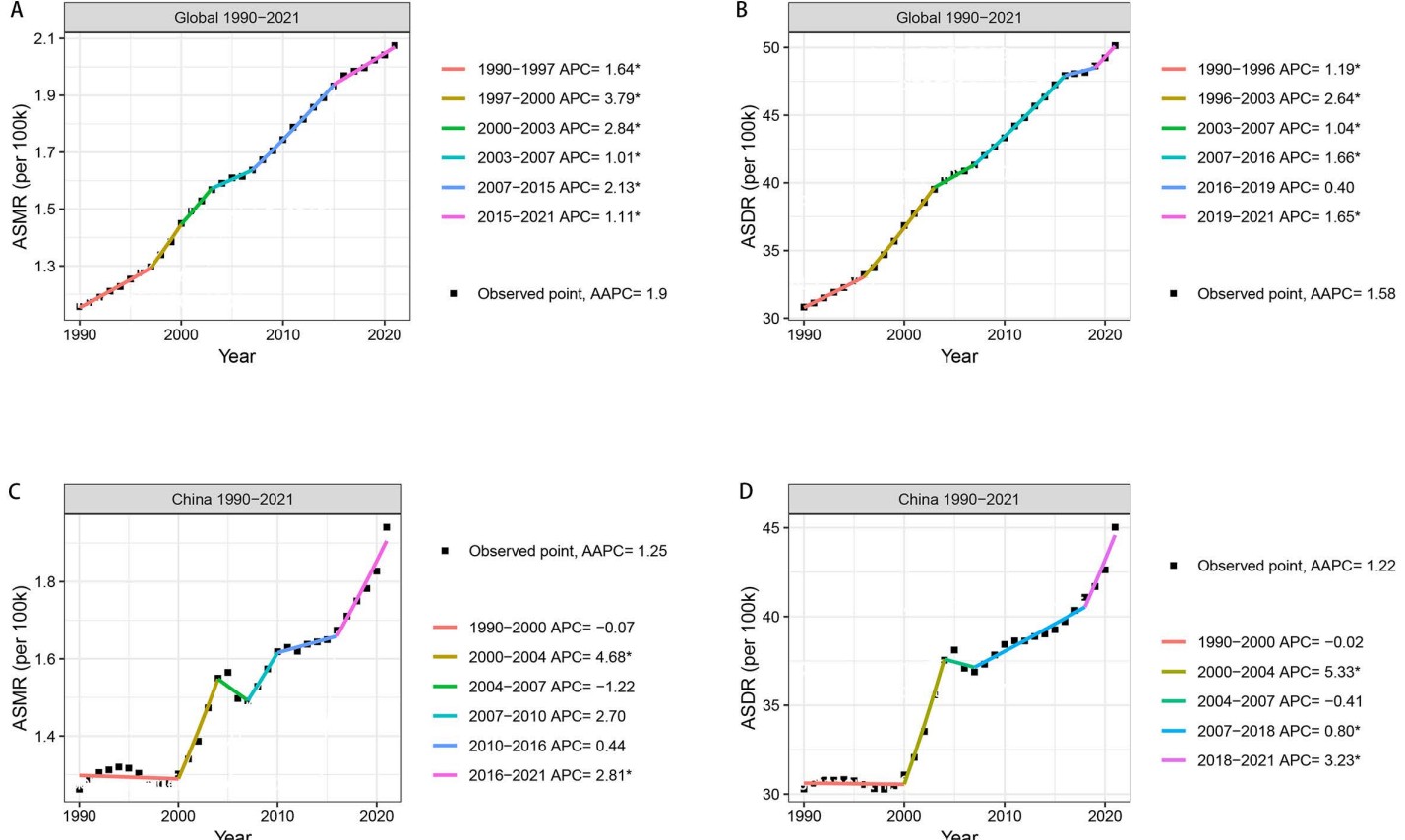

**Fig 3. Annual percentage changes in global and Chinese ASMR and ASDR of DKD attributable to high BMI from 1990 to 2021. (A)** Annual percentage change in global ASMR. **(B)** Annual percentage change in global ASDR. **(C)** Annual percentage change in Chinese ASMR. **(D)** Annual percentage change in Chinese ASDR.

individuals), whereas Northern Europe recorded the lowest (<0.64 per 100,000 individuals). In China in 1990, the ASMR ranged from 1.13 to <1.89 per 100,000 individuals (Fig 4A). By 2021, the global maximum ASMR of DKD attributable to high BMI had increased to below 26.00 per 100,000 individuals. The Caribbean and Central America and the Persian Gulf region continued to lead (2.82 to <26.00 per 100,000 individuals), and Northern Europe continued to record the lowest ASMR (<0.92 per 100,000 individuals). In China in 2021, the ASMR ranged from 1.62 to <2.82 per 100,000 individuals (Fig 4B). In 1990, the global maximum ASDR was below 253.03 per 100,000 individuals. The Caribbean and Central America and the Persian Gulf region had the highest ASDR (48.50 to <253.03 per 100,000 individuals), whereas Northern Europe had the lowest (<21.76 per 100,000 individuals). In China in 1990, the ASDR was between 29.85 and <48.50 per 100,000 individuals (Fig 5A). By 2021, the global maximum ASDR had risen to below 565.86 per 100,000 individuals. The Caribbean and Central America and the Persian Gulf region continued to lead in the rankings for ASDR (67.26 to <565.86 per 100,000 individuals), and Northern Europe continued to record the lowest rates (mostly <24.75 per 100,000 individuals). In China in 2021, the ASDR ranged from 41.81 to <67.26 per 100,000 individuals (Fig 5B).

### 3.5. Relationship between SDI and ASDR as well as ASMR

Globally, the SDI associated with DKD attributable to high BMI was negatively correlated with the overall distribution of ASMR and ASDR. In countries with SDIs ranging from 0.5 to 0.6, such as many African countries, both ASMR and ASDR

A

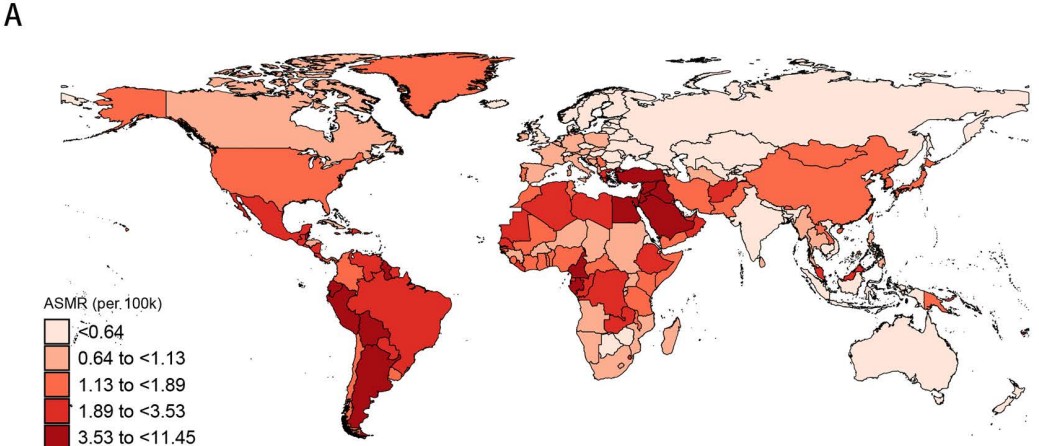

ASMR (per 100k)
<0.64
0.64 to <1.13
1.13 to <1.89
1.89 to <3.53
3.53 to <11.45

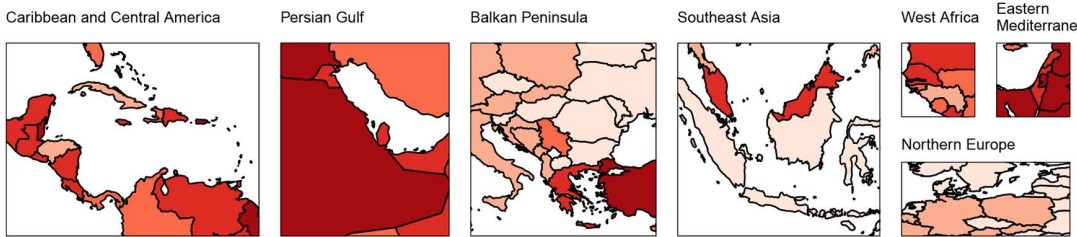

Caribbean and Central America    Persian Gulf    Balkan Peninsula    Southeast Asia    West Africa    Eastern Mediterranea

Northern Europe

B

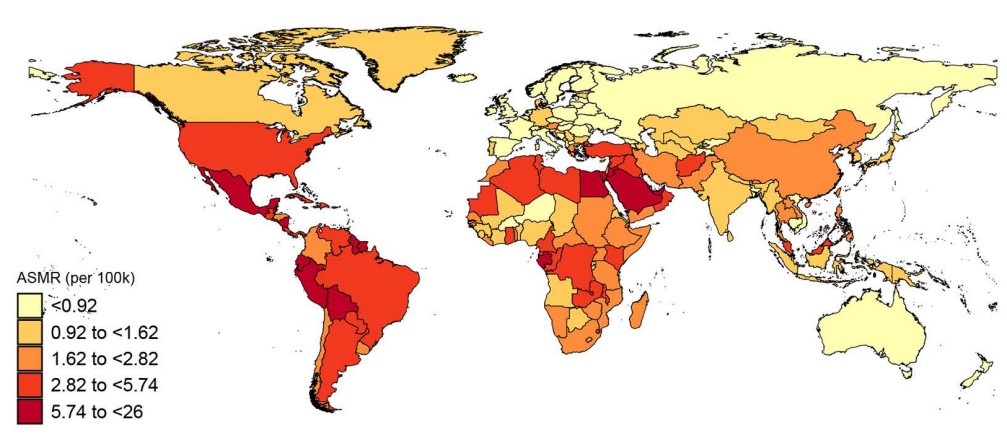

ASMR (per 100k)
<0.92
0.92 to <1.62
1.62 to <2.82
2.82 to <5.74
5.74 to <26

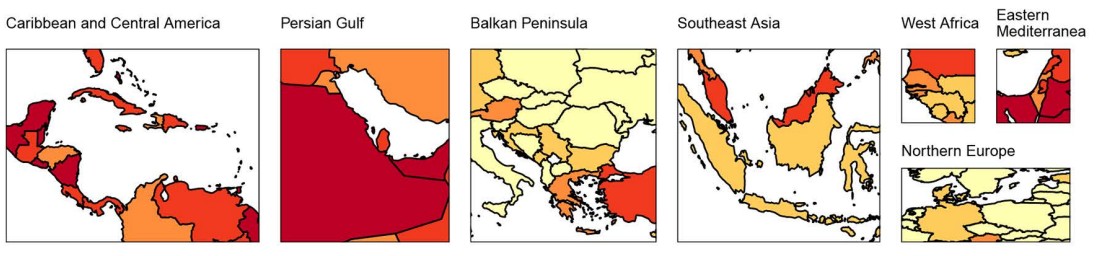

Caribbean and Central America    Persian Gulf    Balkan Peninsula    Southeast Asia    West Africa    Eastern Mediterranea

Northern Europe

**Fig 4. Spatiotemporal distribution of the ASMR of DKD attributable to high BMI across different regions of the world in 1990 and 2021. (A)** In 1990, China was in a lower-burden category. **(B)** By 2021, China had progressed into a higher-burden category, demonstrating a significant increase in its relative burden on the world stage.

increased with rising SDIs. Conversely, in countries with SDIs greater than 0.75, including those in Northern Europe, ASMR and ASDR typically declined with increasing SDIs. Countries with SDIs between 0.60 and 0.75 exhibited relatively high and concentrated ASMRs and ASDRs. However, a few deviants have been identified both overall and locally. American Samoa, a country with a moderately high SDI (0.720), significantly deviates from the general trend expected on the basis of its SDI, exhibiting unusually high ASMRs and ASDRs. China, with an upper-middle SDI (0.722), has ASMR and ASDR values approaching those of developed countries. The correlation coefficient (r) for ASMR was calculated to be −0.19 with a P-value of 0.006, and that for ASDR was calculated to be −0.22 with a P-value of 0.001 (Fig 6A and 6B).

### 3.6. Prediction of the DKD burden attributable to high BMI by 2045 in China and globally

The burden of DKD attributable to high BMI globally and in China from 2022 to 2045 was forecasted using the NORD-PRED model. By 2029, the number of deaths associated with DKD attributable to high BMI is forecasted to rise to 233,204 in the world, with an ASMR of 4.02 per 100,000 individuals, and to 51,331 in China, with an ASMR of 3.63 per 100,000 individuals. By 2045, the death toll is anticipated to further escalate to 361,681 in the world, with an ASMR of 3.90 per 100,000 individuals, and to 84,631 in China, with an ASMR of 3.45 per 100,000 individuals (Fig 7A). By 2029, the DALYs associated with DKD attributable to high BMI are predicted to reach 5.5178 million person-years in the world, with an ASDR of 94.79 per 100,000 individuals, and 1.2064 million person-years in China, with an ASDR of 83.84 per 100,000 individuals. By 2045, the DALYs are projected to surge to 7.6168 million person-years in the world, with an ASDR of 91.54 per 100,000 individuals, and to 1.6539 million person-years in China, with an ASDR of 80.16 per 100,000 individuals (Fig 7B).

## 4. Discussion

The principal contribution of this study is not to reiterate the rising global burden of high BMI-attributable kidney disease, a trend already established by recent analyses [16,17]. Instead, our work provides a novel, focused assessment of this escalating crisis within the Chinese population. By positioning China's data against global benchmarks, we reveal a concerning and divergent trajectory that carries profound implications for national public health policy.

Our most striking finding is the dramatic acceleration of the disease burden in China. From 1990 to 2021, the number of deaths attributable to high BMI-DKD in China surged by 355%, a rate of increase far exceeding the global figure. This "widening gap" suggests that while high BMI is a worldwide issue, its downstream consequences are materializing with exceptional speed and severity in China. This is likely driven by the nation's unique and compressed transition, characterized by rapid urbanization and a shift to unhealthy lifestyles over the past three decades [18–21]. Our joinpoint analysis pinpoints specific periods of accelerated growth (e.g., 2000–2004), which may correspond to key phases in China's socioeconomic development and subsequent increases in DKD detection rates following healthcare reforms [22,23].

Furthermore, our study uncovers a unique demographic vulnerability: the disproportionately high mortality and disability burden among men in China aged 85 and above. This finding, not mirrored in the global aggregate, represents a key novelty of our analysis. It points to a confluence of factors specific to this cohort, where traditional cultural roles may place greater life pressures on men, leading to less healthy lifestyles, which, combined with the cumulative effects of high BMI, results in a higher disease burden in old age [24]. This suggests that generic prevention strategies are insufficient; gender- and age-specific interventions are critically needed for this high-risk group in China.

A

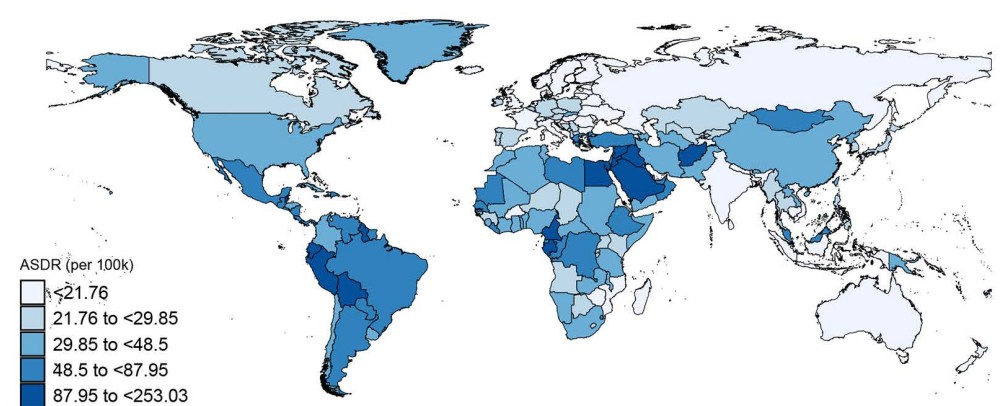

ASDR (per 100k)
<21.76
21.76 to <29.85
29.85 to <48.5
48.5 to <87.95
87.95 to <253.03

Caribbean and Central America    Persian Gulf    Balkan Peninsula    Southeast Asia    West Africa    Eastern Mediterranea

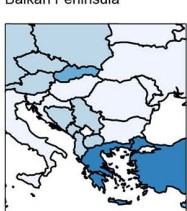

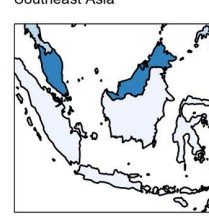

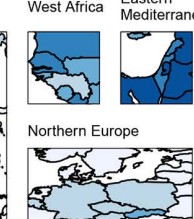

Northern Europe

B

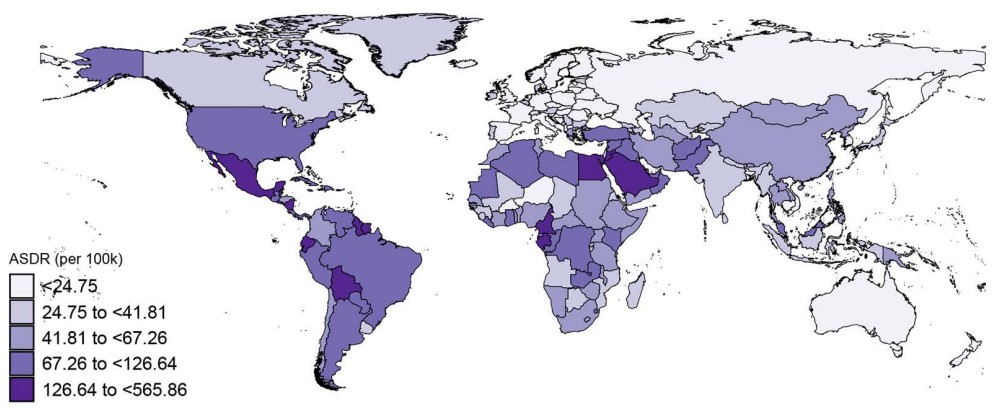

ASDR (per 100k)
<24.75
24.75 to <41.81
41.81 to <67.26
67.26 to <126.64
126.64 to <565.86

Caribbean and Central America    Persian Gulf    Balkan Peninsula    Southeast Asia    West Africa    Eastern Mediterranea

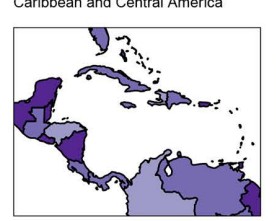

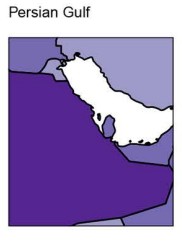

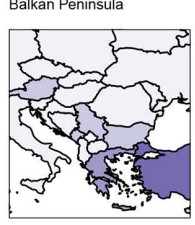

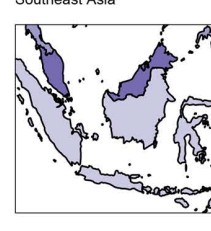

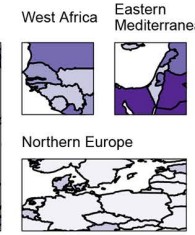

Northern Europe

**Fig 5. Spatiotemporal distribution of the ASDR of DKD attributable to high BMI across different regions of the world in 1990 and 2021. (A)** In 1990, China was in a lower-burden category. **(B)** By 2021, China had progressed into a higher-burden category, demonstrating a significant increase in its relative burden on the world stage.

The policy relevance of these findings is immediate. The ambitious goals of the "Healthy China 2030" initiative are directly threatened by the trajectory our forecasts reveal [19]. Our projections, which predict the number of deaths to more than double by 2045, serve as a stark, data-driven call to action. The SDI analysis further reinforces this, showing that despite its progress to an upper-middle SDI, China has not yet "bent the curve" on this metabolic disease burden, unlike many high-SDI nations where rates have stabilized or declined [25].

### 4.1. Future perspectives and integrated strategies

While this study isolates the impact of high BMI, it is crucial to recognize that in clinical practice, obesity rarely exists in isolation. It frequently clusters with other metabolic risk factors, including hypertension, hyperglycemia, and dyslipidemia, which act synergistically to accelerate DKD progression [26,27]. Therefore, an effective response must embed BMI control within a comprehensive cardiometabolic risk management framework. Future strategies should leverage recent technological, pharmacological, and mechanistic advances. For instance, digital tools like the AWARE web application can facilitate scalable, early identification of cardiovascular and potential renal risk in high-BMI patients [28]. On the pharmacological front, compelling real-world evidence shows that combining GLP-1 receptor agonists (GLP-1 RAs) with SGLT2 inhibitors significantly improves both glycemic and weight control in high-risk obese individuals [29]. This approach, alongside intensive cardiometabolic management proven to induce atherosclerotic plaque regression [30], represents a powerful multi-faceted intervention. Furthermore, emerging mechanistic insights, such as the immunomodulatory benefits of GLP-1 RA activation, may offer novel therapeutic targets within DKD pathophysiology [31]. The strategic integration of these advances is paramount to curbing the rising DKD burden.

### 4.2. Limitations

Our study has several limitations that warrant acknowledgment. First, its reliance on modeled data from the GBD study introduces potential variability in data quality and availability across countries, which may influence the accuracy of the burden estimates. Second, as a study focused on attributable burden, our analysis by design isolates the effect of high BMI and does not quantify the complex interplay with other synergistic risk factors. Future research using prospective, patient-level data is needed to validate our findings and to more deeply explore the interactive effects of multiple metabolic factors on DKD risk, particularly in high-burden regions like China.

### 5. Conclusion

In conclusion, by moving beyond a descriptive global overview, our analysis provides a focused, predictive, and policy-relevant assessment of the high BMI-attributable DKD epidemic in a nation critical to global health. The burden in China is not only increasing but is doing so at an alarming rate and with unique demographic vulnerabilities. These findings issue an urgent call for targeted, evidence-based national strategies– incorporating both population-level prevention and advanced clinical management– to mitigate a formidable and fast-approaching public health crisis.

### Acknowledgments

We sincerely appreciate the efforts of the collaborators involved in the 2021 Global Burden of Disease Study, who have worked to deliver the most comprehensive research on various diseases worldwide.

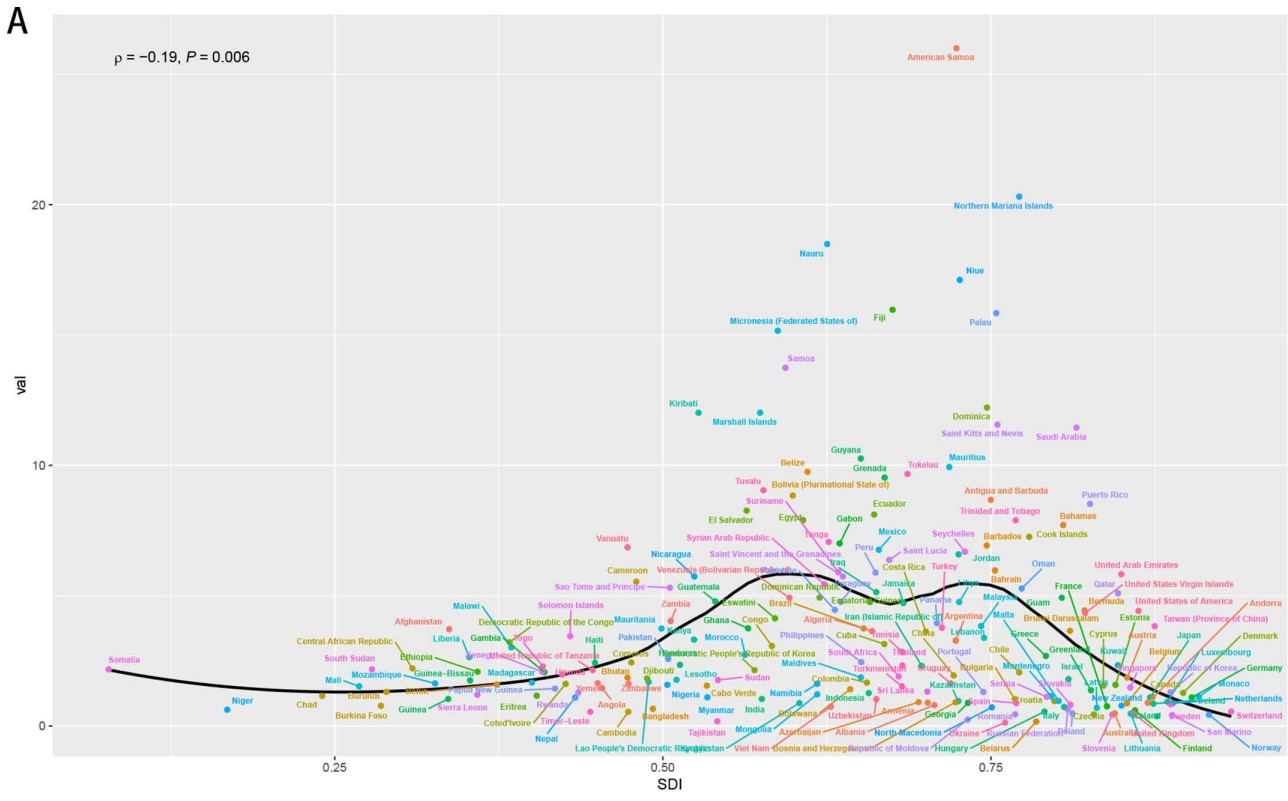

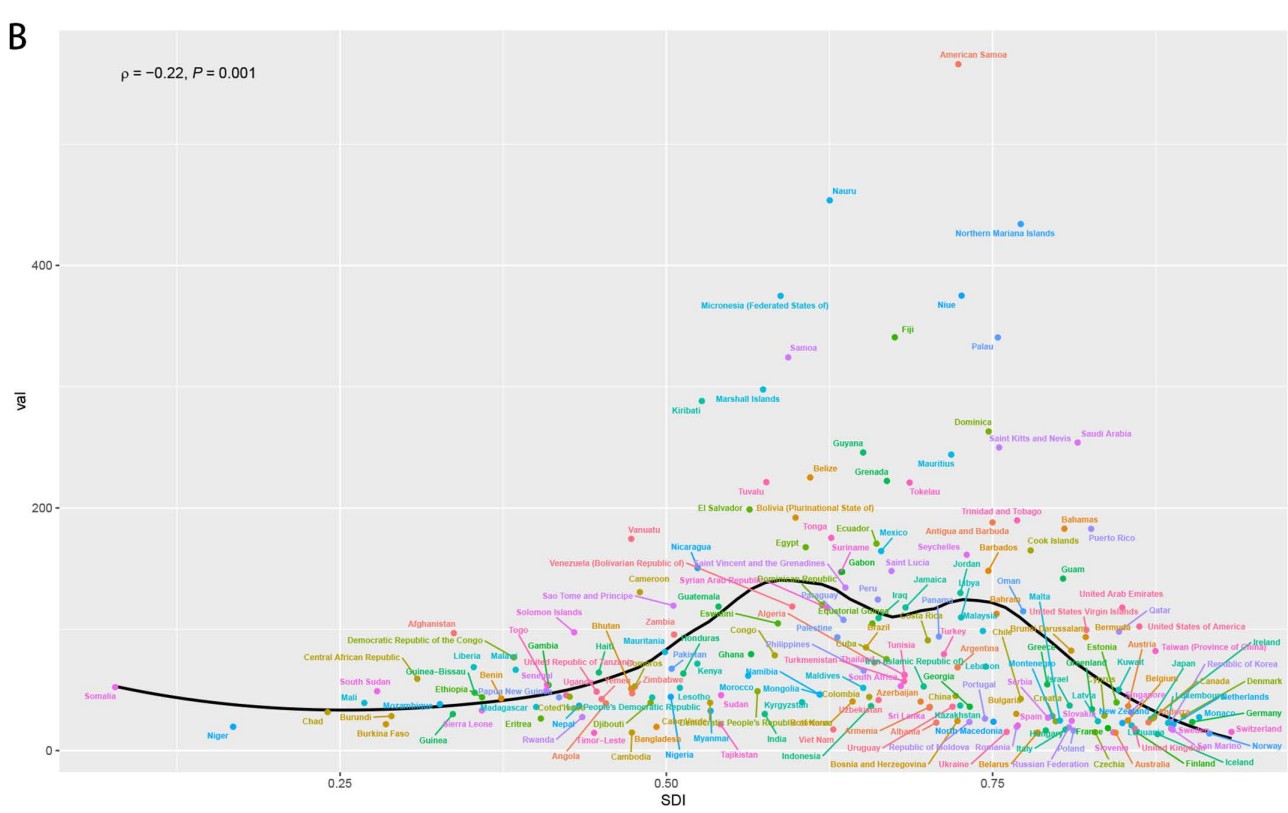

**Fig 6. Global regional distribution of SDI in relation to ASMR and ASDR of DKD attributable to high BMI. (A)** Distribution of SDI in relation to ASMR. **(B)** Distribution of SDI in relation to ASDR. SDI, sociodemographic index.

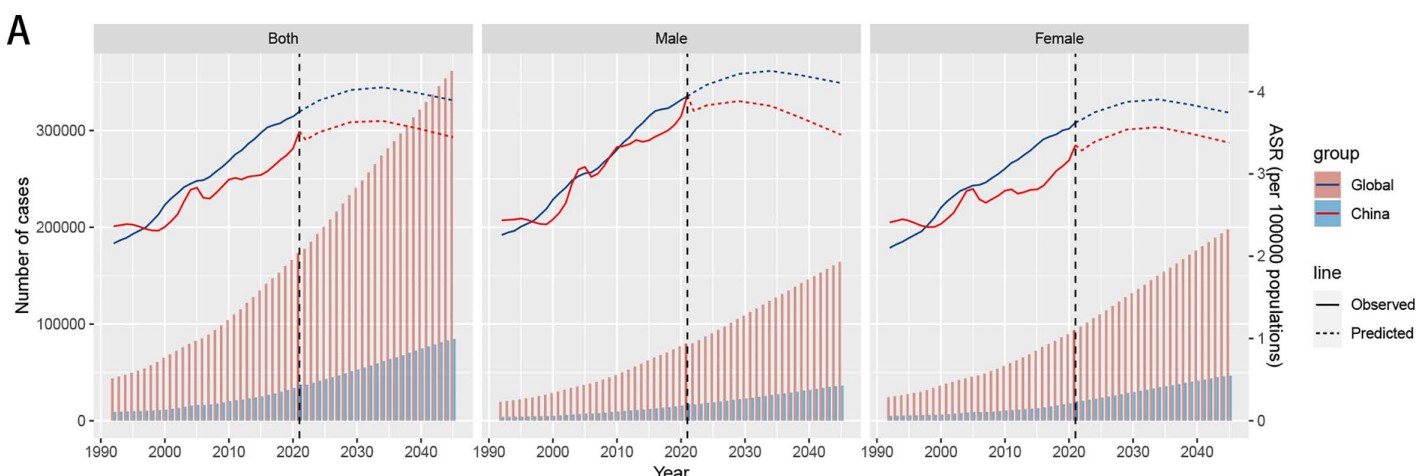

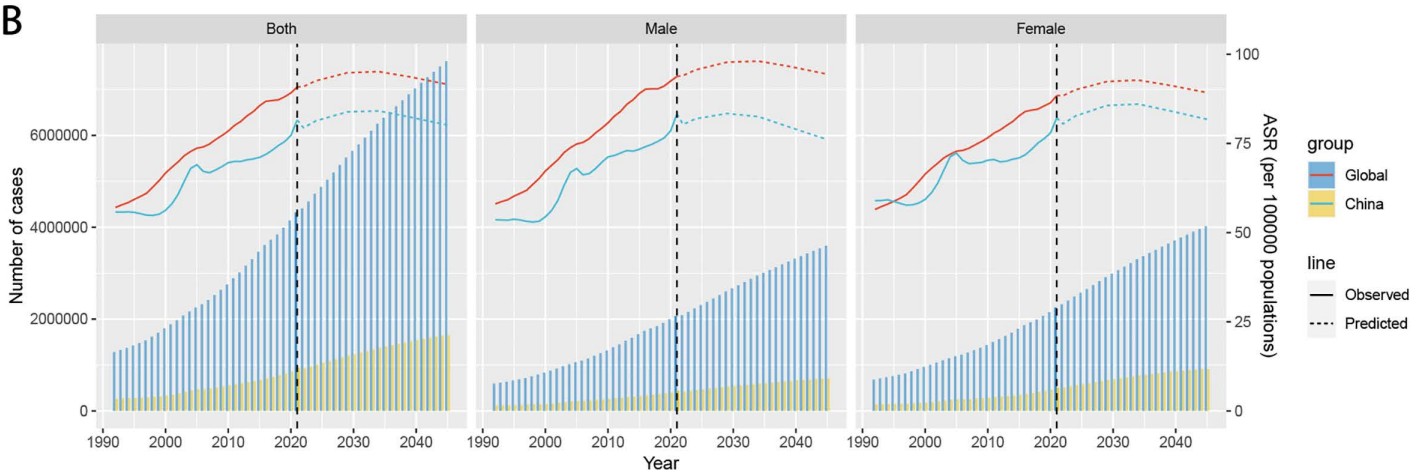

**Fig 7. Predictive analysis of the burden of diabetic kidney disease attributable to high BMI globally and in China for the period of 2022–2045. (A)** Projections of number of deaths and ASMR globally and in China. **(B)** Projections of DALYs and ASDR globally and in China.

## Author contributions

**Conceptualization:** Luyao Jiang, Tianxi Chen.

**Data curation:** Luyao Jiang, Tianxi Chen, Linlin Xu.

**Formal analysis:** Luyao Jiang, Jinkang Jia, Tianxi Chen.

**Methodology:** Jinkang Jia, Tianxi Chen.

**Software:** Tianxi Chen, Linlin Xu.

**Supervision:** Tianxi Chen, Miaojun Shi.

**Visualization:** Luyao Jiang, Tianxi Chen.

**Writing – original draft:** Luyao Jiang.

**Writing – review & editing:** Luyao Jiang, Tianxi Chen.

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
