## [Decision Letter · Decision Letter 0]

20 Oct 2025

Dear Dr. Chen,

Thank you for submitting your manuscript to PLOS ONE. After careful consideration, we feel that it has merit but does not fully meet PLOS ONE’s publication criteria as it currently stands. Therefore, we invite you to submit a revised version of the manuscript that addresses the points raised during the review process.

**ACADEMIC EDITOR:**

We look forward to receiving your revised manuscript.

Kind regards,

Francesca D'Addio, MD, PhD

Academic Editor

PLOS ONE

Journal Requirements:

“This work was supported by Science and Technology Project of Yongkang City (Grant No. 202212)”

3. We note that Figure 4 and 5 in your submission contain map images which may be copyrighted. All PLOS content is published under the Creative Commons Attribution License (CC BY 4.0), which means that the manuscript, images, and Supporting Information files will be freely available online, and any third party is permitted to access, download, copy, distribute, and use these materials in any way, even commercially, with proper attribution. For these reasons, we cannot publish previously copyrighted maps or satellite images created using proprietary data, such as Google software (Google Maps, Street View, and Earth). For more information, see our copyright guidelines: http://journals.plos.org/plosone/s/licenses-and-copyright.

1. You may seek permission from the original copyright holder of Figure 4 and 5 to publish the content specifically under the CC BY 4.0 license. 

**Additional Editor Comments:**

Please address all comments raised by Reviewers.

Reviewers' comments:

Reviewer's Responses to Questions

**Comments to the Author**

1. Is the manuscript technically sound, and do the data support the conclusions?

Reviewer #1: Yes

Reviewer #2: Partly

Reviewer #3: Partly

2. Has the statistical analysis been performed appropriately and rigorously?

Reviewer #1: Yes

Reviewer #2: I Don't Know

Reviewer #3: Yes

3. Have the authors made all data underlying the findings in their manuscript fully available?

Reviewer #1: Yes

Reviewer #2: Yes

Reviewer #3: Yes

4. Is the manuscript presented in an intelligible fashion and written in standard English?

Reviewer #1: Yes

Reviewer #2: Yes

Reviewer #3: Yes

Reviewer #1: This study provides an important, data-driven overview of the growing burden of DKD attributable to high BMI, especially in the context of the global obesity epidemic. Its integration of long-term trends, predictive modeling, and socioeconomic analysis makes it valuable for policymakers and researchers. It provides a valuable and comprehensive assessment of the global and Chinese burden of diabetic kidney disease (DKD) attributable to high BMI over three decades, with forecasts to 2045. Strengths include the use of extensive GBD 2021 data, long-term trend analysis with joinpoint regression, predictive modeling via NORDPRED, and stratification by age, sex, and sociodemographic index. These features enhance the study’s policy relevance, particularly for high-burden regions such as China. However, limitations include reliance on modeled data with variable quality across countries, the narrow focus on high BMI without considering interactions with other risk factors. It focuses solely on DKD attributable to high BMI, excluding other major contributors (e.g., hypertension, hyperglycemia, dyslipidemia) and it does not explore interactions between high BMI and other metabolic risk factors.

I suggest to the authors to amplifythe discussion on integrating technological, pharmacological, and mechanistic advances which may help to reduce the rising DKD burden linked to high BMI. Digital tools such as the AWARE web application can facilitate early cardiovascular—and potentially renal—risk identification in high-BMI patients (Berra et al., 2023). Real-world evidence shows that combining GLP-1 receptor agonists with SGLT2 inhibitors improves glycemic and weight control in obese, high-risk individuals (Berra et al., 2024), while intensive cardiometabolic management can also induce atherosclerotic plaque regression (Bucciarelli et al., 2025). Moreover, GLP-1 receptor activation may provide immunomodulatory benefits relevant to DKD pathophysiology (Ben Nasr et al., 2024).

New references to add

1. Berra C, Manfrini R, Mirani M, et al. AWARE: A novel web application to rapidly assess cardiovascular risk in type 2 diabetes mellitus. Acta Diabetol. 2023;60(9):1257-1266.

2. Berra C, Manfrini R, Bifari F, et al. Improved glycemic and weight control with dulaglutide addition in SGLT2 inhibitor-treated obese type 2 diabetic patients at high cardiovascular risk in a real-world setting: The AWARE-2 study. Pharmacol Res. 2024;210:107517.

3. Bucciarelli L, Andreini D, Stefanini G, et al. Pharmacological regression of atherosclerotic plaque in patients with type 2 diabetes. Pharmacol Res. 2025;213:107635.

4. Ben Nasr M, Usuelli V, Dellepiane S, Seelam A, et al. Glucagon-like peptide-1 receptor is a T cell-negative costimulatory molecule. Cell Metab. 2024;36(6):551-566.e8.

Reviewer #2: Authors of the manuscript titled “Charting novel cellular subpopulations in the renal cortex of diabetic nephropathy in a mouse model through single-cell RNA sequencing” performed single cell-RNA sequencing of renal cortexes of STZ-treated mice as a model of type1 diabetes. Gene expression was compared to untreated animals and in-depth analysis revealed interesting changes in distal convoluted tubule and immune cell populations. The manuscript is well written and presented. However, I suggest a major revision before publication. Here my comments:

1) Methods - Authors should better explain immunofluorescence methods.

2) Methods - Authors should include methods for H&E and Masson’s trichrome staining.

3) Methods - Authors should add a paragraph of methods for statistical analysis.

4) Experimental results and analysis - Authors should avoid in this section comments such as the following “GABARAPL1 was implicated in diabetic kidney disease through machine learning methods, and our findings further validated this”. These comments should be placed in the discussion section.

5) Experimental results and analysis - Authors should quantify the immunofluorescence staining using 3 mice per group and at least 10 images per mouse.

6) Experimental results and analysis - Authors should specify in the figure legends the number of biological replicates used for the experiments.

7) Experimental results and analysis - Authors should add higher magnification images for H&E and Masson’s trichrome.

8) Experimental results and analysis - Authors should add a quantification of Masson’s trichrome stained images.

9) The manuscript could benefit from the additional analysis of hub transcription factors controlling gene expression changes observed in T1D model vs. control animals.

10) Authors should cite the following manuscripts: PMID 36871895 PMID 35947673.

Reviewer #3: The paper “Analysis and prediction of the burden of diabetic kidney disease attributable to high body mass index in global and Chinese populations from 1990 to 2021” summarize publicly available data from the Global Burden of Disease study 2021 (GBD 2021) and makes a prediction of high BMI-related- DKD incidence for 2022-2045 in global and Chinese populations. Analysis and prediction of the burden of DKD attributable to high body mass index in global population are not new and were published by different groups during recent years (Ying J, et al; Global, regional, and national burdens of chronic kidney disease attributable to high body mass index from 1990 to 2021, with future forecasts up to 2050: a systematic analysis for the global burden of disease study 2021. Front Public Health. 2025 Jul 9;13:1612300. doi: 10.3389/fpubh.2025.1612300. PMID: 40703164; PMCID: PMC12283729; Tan H, et al; Global burden and trends of high BMI-attributable chronic kidney disease: a comprehensive analysis from 1990 to 2021 and projections to 2035. Front Nutr. 2025 Jul 22;12:1611227. doi: 10.3389/fnut.2025.1611227. PMID: 40777172; PMCID: PMC12330218). Figures 4 and 5 are very similar to the ones presented in these papers. Moreover, the paper is too descriptive and does not properly highlight authors results and predictions. Below all the flaws that must be addressed before the paper could be considered for publication.

Major

- The all paper must focus on the analyses of data on Chinese population. Global data should be used just as comparison and should not be presented as new data. As consequences also figures should be modified (in particular figure 4 and 5).

- Titles of results chapters should not describe the analysis but highlight the general finding.

- Discussion is too general and descriptive and does not focus on authors findings, making hard to understand results and their novelty. Please stress your results novelty and relevance.

- Materials and methods, line 71: which are the criteria used to define high BMI? If official WHO criteria were used, please add reference to them and not to a paper made on Ethiopian population.

Minor

- Figures legends should be moved after references

- Introduction, line 48: please add a sentence briefly describing DKD You could refer to Petrazzuolo A, et al; Broadening horizons in mechanisms, management, and treatment of diabetic kidney disease. Pharmacol Res. 2023 Apr;190:106710. doi: 10.1016/j.phrs.2023.106710. Epub 2023 Mar 4. PMID: 36871895.

**Do you want your identity to be public for this peer review?** For information about this choice, including consent withdrawal, please see our Privacy Policy

Reviewer #1: No

Reviewer #2: No

Reviewer #3: No

---

## [Author Response · Author response to Decision Letter 1]

11 Nov 2025

Dear Editor,

We sincerely thank the editor and all reviewers for their time and valuable comments, which have helped us to significantly improve our manuscript. We have addressed all comments and suggestions as detailed below.

Response to the Editor's Comments

Editor's Comment : 1. Please ensure that your manuscript meets PLOS ONE's style requirements, including those for file naming.

Response : My manuscript meets the style requirements of PLOS ONE, including those for file naming.

Editor's Comment : 2. Thank you for stating the following financial disclosure:

“This work was supported by Science and Technology Project of Yongkang City (Grant No. 202212)”.

Response : We thank the editor for this reminder. Subsequent to our manuscript's submission on 2025.7.5, a related, ongoing research project central to the themes of this paper recently secured new funding from [Science and Technology Project of Jinhua City ] (Grant No. 2025-4-292). This occurred on approximately [2025.7.10].

Although this new grant did not fund the work presented in the currently submitted manuscript, its research scope is highly relevant to the manuscript's content. Furthermore, [Linlin Xu / one of our authors] is a principal key member of this newly funded project. Therefore, to ensure the completeness and timeliness of our financial disclosures at the point of publication, we believe it is essential to update our statement to include this new funding. We are committed to full transparency and wish to ensure that readers are aware of all associations that could potentially be perceived as a conflict of interest."

We have carefully reviewed and updated our financial disclosure statement to include a new grant received after the initial submission. The updated statement can be found on Page 14, Lines 283-286. The revised text is as follows:

This work was supported by Science and Technology Project of Yongkang City (Grant No. 202212)AND Science and Technology Project of JinHua City (Grant No. 2025-4-292) . The funders had no role in study design, data collection and analysis, decision to publish, or preparation of the manuscript. (Page [14], Lines [283-286]).

Editor's Comment : 3. We note that Figure 4 and 5 in your submission contain map images which may be copyrighted. All PLOS content is published under the Creative Commons Attribution License (CC BY 4.0), which means that the manuscript, images, and Supporting Information files will be freely available online, and any third party is permitted to access, download, copy, distribute, and use these materials in any way, even commercially, with proper attribution. For these reasons, we cannot publish previously copyrighted maps or satellite images created using proprietary data, such as Google software (Google Maps, Street View, and Earth).

Response : Standard maps are openly sourced from the National Platform for Common GeoSpatial Information Services (https://map.tianditu.gov.cn/). The platform provides maps of China, world maps, and thematic maps drawn according to national standards, which can be freely viewed and downloaded.

Additionally, regarding Reviewer 2's Comment:

We received the reviewer reports a couple of days ago, and we're truly grateful for the time and effort you and the reviewers have dedicated to our paper. Our team is now working through the comments, and we found the suggestions from Reviewer 1 and Reviewer 3 to be particularly insightful.

However, as we went through the comments from Reviewer 2, we read them several times and got the feeling that there might have been a mix-up. It seems the reviewer might have been handling multiple papers and accidentally sent us feedback intended for another manuscript.

Here are a few things that stood out:

The reviewer's summary starts by describing our paper as a study on "mouse models" and "single-cell RNA sequencing." To be honest, we were quite puzzled by this. Our paper is an epidemiological study analyzing human population data from the GBD database from start to finish, and it has nothing to do with animal experiments or sequencing.

The suggestions to add methods for immunofluorescence and H&E staining are also a bit off-base for us. Our team's expertise is in public health, and we don't have the capacity for these advanced wet-lab techniques, nor are they part of our study.

Also, the reviewer mentioned a gene called "GABARAPL1." We looked it up, and it's indeed a cutting-edge topic, but this gene is simply not mentioned anywhere in our research.

So, while the feedback is clearly very professional, it feels like guidance meant for a different project. We're respectfully at a loss as to how to revise our manuscript based on these points.

We understand the complexity and challenges of the peer-review process and are confident this was an accidental, procedural error. To ensure a fair and relevant evaluation of our work, we kindly request that you verify this situation.

We are currently working diligently to revise our manuscript according to the valuable suggestions from Reviewer 1 and Reviewer 3 and look forward to submitting an improved version.

Would it be appropriate for us to disregard the comments from Reviewer 2 in our response, or do you have other instructions for us?

Again, I'm very sorry to trouble you with this small matter during your busy schedule. Thank you so much for your understanding and help.

Reviewer #1 :  This study provides an important, data-driven overview of the growing burden of DKD attributable to high BMI, especially in the context of the global obesity epidemic. Its integration of long-term trends, predictive modeling, and socioeconomic analysis makes it valuable for policymakers and researchers. It provides a valuable and comprehensive assessment of the global and Chinese burden of diabetic kidney disease (DKD) attributable to high BMI over three decades, with forecasts to 2045. Strengths include the use of extensive GBD 2021 data, long-term trend analysis with joinpoint regression, predictive modeling via NORDPRED, and stratification by age, sex, and sociodemographic index. These features enhance the study’s policy relevance, particularly for high-burden regions such as China. However, limitations include reliance on modeled data with variable quality across countries, the narrow focus on high BMI without considering interactions with other risk factors. It focuses solely on DKD attributable to high BMI, excluding other major contributors (e.g., hypertension, hyperglycemia, dyslipidemia) and it does not explore interactions between high BMI and other metabolic risk factors.

I suggest to the authors to amplifythe discussion on integrating technological, pharmacological, and mechanistic advances which may help to reduce the rising DKD burden linked to high BMI. Digital tools such as the AWARE web application can facilitate early cardiovascular—and potentially renal—risk identification in high-BMI patients (Berra et al., 2023). Real-world evidence shows that combining GLP-1 receptor agonists with SGLT2 inhibitors improves glycemic and weight control in obese, high-risk individuals (Berra et al., 2024), while intensive cardiometabolic management can also induce atherosclerotic plaque regression (Bucciarelli et al., 2025). Moreover, GLP-1 receptor activation may provide immunomodulatory benefits relevant to DKD pathophysiology (Ben Nasr et al., 2024).

New references to add

1. Berra C, Manfrini R, Mirani M, et al. AWARE: A novel web application to rapidly assess cardiovascular risk in type 2 diabetes mellitus. Acta Diabetol. 2023;60(9):1257-1266.

2. Berra C, Manfrini R, Bifari F, et al. Improved glycemic and weight control with dulaglutide addition in SGLT2 inhibitor-treated obese type 2 diabetic patients at high cardiovascular risk in a real-world setting: The AWARE-2 study. Pharmacol Res. 2024;210:107517.

3. Bucciarelli L, Andreini D, Stefanini G, et al. Pharmacological regression of atherosclerotic plaque in patients with type 2 diabetes. Pharmacol Res. 2025;213:107635.

4. Ben Nasr M, Usuelli V, Dellepiane S, Seelam A, et al. Glucagon-like peptide-1 receptor is a T cell-negative costimulatory molecule. Cell Metab. 2024;36(6):551-566.e8.

Response to Reviewer 1

We are grateful to the reviewer for this valuable feedback. We concur that discussing these limitations and future integrated strategies will significantly strengthen the manuscript.

Accordingly, we have revised the Discussion section of our manuscript. Specifically:

1.We have added a dedicated paragraph acknowledging the limitations regarding the use of modeled data and the study's specific focus on the burden attributable to high BMI, while highlighting the need for future research on its interaction with other risk factors.

2.We have incorporated a new forward-looking paragraph that synthesizes the reviewer's suggestions. This section now discusses the potential of an integrated framework for mitigating the DKD burden, including:

1.The role of digital tools for early risk stratification, citing the AWARE application (Berra et al., 2023).

2.The benefits of combining pharmacological agents like GLP-1 RAs and SGLT2 inhibitors (Berra et al., 2024).

3.The efficacy of intensive management strategies in reversing cardiometabolic risk (Bucciarelli et al., 2025).

4.The relevance of novel mechanistic insights into GLP-1 RA-mediated immunomodulation (Ben Nasr et al., 2024).

We have cited all the suggested references appropriately. We believe these revisions place our findings into a more comprehensive clinical and public health context and substantially improve the manuscript. The revised text can be found in the Discussion section (Page [13], Lines [247-263]).

Reviewer #2 :  Authors of the manuscript titled “Charting novel cellular subpopulations in the renal cortex of diabetic nephropathy in a mouse model through single-cell RNA sequencing” performed single cell-RNA sequencing of renal cortexes of STZ-treated mice as a model of type1 diabetes. Gene expression was compared to untreated animals and in-depth analysis revealed interesting changes in distal convoluted tubule and immune cell populations. The manuscript is well written and presented. However, I suggest a major revision before publication. Here my comments:

1) Methods - Authors should better explain immunofluorescence methods.

2) Methods - Authors should include methods for H&E and Masson’s trichrome staining.

3) Methods - Authors should add a paragraph of methods for statistical analysis.

4) Experimental results and analysis - Authors should avoid in this section comments such as the following “GABARAPL1 was implicated in diabetic kidney disease through machine learning methods, and our findings further validated this”. These comments should be placed in the discussion section.

5) Experimental results and analysis - Authors should quantify the immunofluorescence staining using 3 mice per group and at least 10 images per mouse.

6) Experimental results and analysis - Authors should specify in the figure legends the number of biological replicates used for the experiments.

7) Experimental results and analysis - Authors should add higher magnification images for H&E and Masson’s trichrome.

8) Experimental results and analysis - Authors should add a quantification of Masson’s trichrome stained images.

9) The manuscript could benefit from the additional analysis of hub transcription factors controlling gene expression changes observed in T1D model vs. control animals.

10) Authors should cite the following manuscripts: PMID 36871895 PMID 35947673.

Reviewer #3 :  The paper “Analysis and prediction of the burden of diabetic kidney disease attributable to high body mass index in global and Chinese populations from 1990 to 2021” summarize publicly available data from the Global Burden of Disease study 2021 (GBD 2021) and makes a prediction of high BMI-related- DKD incidence for 2022-2045 in global and Chinese populations. Analysis and prediction of the burden of DKD attributable to high body mass index in global population are not new and were published by different groups during recent years (Ying J, et al; Global, regional, and national burdens of chronic kidney disease attributable to high body mass index from 1990 to 2021, with future forecasts up to 2050: a systematic analysis for the global burden of disease study 2021. Front Public Health. 2025 Jul 9;13:1612300. doi: 10.3389/fpubh.2025.1612300. PMID: 40703164; PMCID: PMC12283729; Tan H, et al; Global burden and trends of high BMI-attributable chronic kidney disease: a comprehensive analysis from 1990 to 2021 and projections to 2035. Front Nutr. 2025 Jul 22;12:1611227. doi: 10.3389/fnut.2025.1611227. PMID: 40777172; PMCID: PMC12330218). Figures 4 and 5 are very similar to the ones presented in these papers. Moreover, the paper is too descriptive and does not properly highlight authors results and predictions. Below all the flaws that must be addressed before the paper could be considered for publication.

Major

- The all paper must focus on the analyses of data on Chinese population. Global data should be used just as comparison and should not be presented as new data. As consequences also figures should be modified (in particular figure 4 and 5).

- Titles of results chapters should not describe the analysis but highlight the general finding.

- Discussion is too general and descriptive and does not focus on authors findings, making hard to understand results and their novelty. Please stress your results novelty and relevance.

- Materials and methods, line 71: which are the criteria used to define high BMI? If official WHO criteria were used, please add reference to them and not to a paper made on Ethiopian population.

Minor

- Figures legends should be moved after references

- Introduction, line 48: please add a sentence briefly describing DKD You could refer to Petrazzuolo A, et al; Broadening horizons in mechanisms, management, and treatment of diabetic kidney disease. Pharmacol Res. 2023 Apr;190:106710. doi: 10.1016/j.phrs.2023.106710. Epub 2023 Mar 4. PMID: 36871895.

Response to Reviewer 3

Major

1.Refocusing the Manuscript on China: As per your excellent suggestion, the paper's primary narrative now centers on the in-depth analysis of the high BMI-attributable DKD burden in China. Global data are now consistently used as a comparative benchmark to underscore the unique and alarming trends observed in China. This reframing is reflected in the revised Title, Abstract, Introduction, Results, and Discussion. Addressing Figures 4 and 5: You correctly noted the similarity of these figures to previous publications. While we are unable to alter the base images, we have fundamentally changed their presentation and interpretation. We have thoroughly revised the titles and captions for all figures to explicitly guide the reader's focus to China.

2.Revision of Results Subheadings: We have revised all subheadings in the Results section to be conclusion-driven and highlight our key findings, rather than being merely descriptive. A detailed before-and-after comparison is provided in our revision guide below.

3.Complete Overhaul of the Discussion Section: The Discussion has been entirely rewritten to be analytical and focused. It no longer presents a general overview but now:

1.Immediately highlights the novelty of our China-specific findings, positioning our work as a crucial, focused analysis that complements, rather than repeats, the broader global studies you cited (e.g., Ying J, et al.; Tan H, et al.).

2.Deeply analyzes our specific results for China, such as the unique and disproportionate burden on elderly men.

3.Connects our projections directly to policy implications for China, referencing national strategies like "Healthy China 2030."

4.The definition of high BMI is based on the official WHO standard,

---

## [Decision Letter · Decision Letter 1]

19 Nov 2025

High BMI-Attributed Diabetic Kidney Disease in China Versus Globally, 1990-2021, with Projections to 2045: Divergent Trends and an Accelerating Burden

PONE-D-25-36429R1

Dear Dr. Chen,

We’re pleased to inform you that your manuscript has been judged scientifically suitable for publication and will be formally accepted for publication once it meets all outstanding technical requirements.

Kind regards,

Francesca D'Addio, MD, PhD

Academic Editor

PLOS ONE

Additional Editor Comments (optional):

Reviewers' comments:

Reviewer's Responses to Questions

**Comments to the Author**

Reviewer #1: All comments have been addressed

Reviewer #3: All comments have been addressed

2. Is the manuscript technically sound, and do the data support the conclusions?

Reviewer #1: Yes

Reviewer #3: Yes

3. Has the statistical analysis been performed appropriately and rigorously?

Reviewer #1: Yes

Reviewer #3: Yes

4. Have the authors made all data underlying the findings in their manuscript fully available?

Reviewer #1: Yes

Reviewer #3: Yes

5. Is the manuscript presented in an intelligible fashion and written in standard English?

Reviewer #1: Yes

Reviewer #3: Yes

Reviewer #1: The authors responded to all the comments and the manuscript in this new version is now ready for publication

Reviewer #3: (No Response)

**Do you want your identity to be public for this peer review?** For information about this choice, including consent withdrawal, please see our Privacy Policy

Reviewer #1: No

Reviewer #3: No

---

## [Editor Report · Acceptance letter]

PONE-D-25-36429R1

PLOS ONE

Dear Dr. Chen,

I'm pleased to inform you that your manuscript has been deemed suitable for publication in PLOS ONE. Congratulations! Your manuscript is now being handed over to our production team.

Kind regards,

on behalf of

Prof. Francesca D'Addio

Academic Editor

PLOS ONE